# OpenReview forum: "Understanding and Detecting File Knowledge Leakage in GPT App Ecosystem"
_ACM.org/TheWebConf/2025/Conference — WWW 2025 Poster_

### Official Review · Reviewer_xahj · 2024-12-01

**Novelty:** 4
**Technical Quality:** 3

**Review:**

PROS:

 - I would like to thank the authors for the submission. The authors investigate a novel problem related to file knowledge leakage within GPTs.
 - The dataset comprises 8,000 of the most popular GPTs across 8 different categories.
 - The paper is well-written and the methodology used is clear.

CONS:

 - The analysis provided is very superficial in terms of what is considered a leakage. There is no categorization of the type and criticality of the information exposed.
 - The threat model is very limited (prompt injection and transmitted data only).
 - The list of potential attack scenarios discussed in Section 6.1 is not very insightful and not specific to file knowledge leakage.
 - The conclusions and recommendations (Section 6.2) are somewhat obvious and do not provide any additional insights to what is already generally known in the industry.

DETAILED COMMENTS:

The authors use "a pre-built file knowledge harmful prompts library to perform prompt injection on these GPTs". However, the analysis is as good as the quality of these harmful prompts. How were they validated and the authors ensured they are effective for the proposed threat scenarios? In particular, how are these prompt level attacks different from traditional ones?

On page 2 the authors mention the examination of the transmitted data using encryption, but I am certain this was not discussed again anywhere in the paper.

The background section is too lengthy. The paper would be more insightful if this space could be used to present more interesting results.

In Section 5.1 you say: "we use the metadata crawled from GPTsApp.io, specifically the FAQs section, which includes a question, i.e., 'Does this GPTs have its own knowledge base?' This question helps us determine whether a GPTs has its own file knowledge". But could be the case that the FAQ section uses a slightly different phrasing or that it does not include an entry about this at all and still the GPT could have knowledge files? Is the methodology used reliable? Furthermore, how do you check how many files the GPT uses?

The granularity of the analysis is superficial. You say that you can record the number of GPTs that leak file knowledge, the number of leaked files, and the corresponding percentage of leakage. What is considered a leakage? What classes of information could indicate a mild or more severe leakage? The analysis is superficial and more in depth insights should be provided, possibly categorizing the type and criticality of the information exposed.

Earlier in the introduction the authors mentioned encryption, but the results presented regarding transmitted data do not mention encryption anymore.

Regarding the attack scenarios (Section 6.1), in order to perform phishing, attackers would have to convince users into using the counterfeit version of GPT, however this is not a GPT specific attack and is no different from any other type of phishing. Other attacks, such as circumventing prompt injection safeguards, although specific to GPT, are still too broad and not particularly new. In general, the list of potential attack scenarios is not insightful and adds very little to the discussion in its current form.

Likewise, recommendations (Section 6.2) such as strengthening access control and preventing prompt injection, although they are important, do not provide any additional insights to what is already generally known in the industry.

Fig. 1 does not contain a lot of useful information and the space could have been better used to provide more technical details and results.

**Questions:**

- The analysis is superficial in terms of what is considered a leakage. Are you able to you categorize the type and criticality of the information exposed?

 - What if the FAQ section uses a slightly different phrasing from 'Does this GPTs have its own knowledge base?' or if it does not include an entry about this at all and still the GPT has knowledge files? Is the methodology used reliable? Furthermore, how do you check how many files the GPT uses?

 - How do you ensure the created prompt library is effective and complete for the proposed threat scenarios? How was this validated?

 - What is the impact of encryption in the transmitted data analysis? This was briefly mentioned in the introduction but it was never addressed by the authors.

**Reviewer Confidence:**

3: The reviewer is confident but not certain that the evaluation is correct

**Scope:**

3: The work is somewhat relevant to the Web and to the track, and is of narrow interest to a sub-community

---

### Official Review · Reviewer_MEA3 · 2024-12-02

**Novelty:** 4
**Technical Quality:** 3

**Review:**

This work conducts the first study on file knowledge leakage within the most popular LLM apps, i.e., GPTs, through injecting harmful prompts during user-GPTs interaction and deciphering network packets. Evaluation results on the top 8K GPTs show file knowledge can be leaked at both two levels, giving insights for both GPT app store and GPT app developers.

Strengthens
- A large-scale, comprehensive study on GPT app file knowledge leakage.
- The paper is well-organized and easy to follow.

Weaknesses
- Potential violation of ethical rules: Given the following concerns, I encourage the authors to revisit their methodology to think through how they handle the ethical implications.
    - The authors state that they will publish the relevant data and inform developers to avoid ethical violations. However, considering the file knowledge data used in this study were “reverse-engineered” from close-sourced GPT apps, without explicit consent from the owners of the proprietary data, publishing them online will be seen as unethical.
    - Moreover, how do the authors plan to reach out to and inform GPT app developers, given that not all GPT apps provide a website/email/personal webpage in their builder information?
    - An explicit disclosure of the GPT app prompt and its GizmoID (in Section 3.2) should be avoided.
- Unclear unique contributions: The insufficient review of prior research makes inadequate support for the claims made in the paper. For example,
    - “A comprehensive top GPTs dataset.” Prior studies have constructed GPT app datasets on a larger scale; they were also released prior to the submission time of WWW 2025 [1,2]. Comparison with existing GPT app datasets is necessary to demonstrate the uniqueness of the dataset in this work.
    - “The limited existing research on GPT prompt injection has largely focused on limited tests targeting only a few GPTs” “To the best of our knowledge, our work is the first large-scale automated detection of GPTs to retrieve their file knowledge.” [2] conducted a similar analysis on a comparative scale.
- The evaluation design is somewhat flawed: How do the authors ensure that the GPT app metadata crawled from [GPTsApp.io](http://gptsapp.io/) is the ground truth for deriving the results in Section 5.1?
- Design of GPTs-Filtor makes the definition of “file knowledge” unclear:
    - Step I: Please list all the prompts used in your harmful prompts library, rather than showing a few examples.
    - As the shell command prompt mentioned in GPTs-Filtor only sends “ls /mnt/data” to GPT apps (authors described this in Section 4 Step I and Figure 9), I am a bit confused about the consistency of the term “file knowledge” throughout the paper. Some harmful prompts request the download link of knowledge files, while others only request file names. Therefore, based on the system design and evaluation results, what I obtained is that “file knowledge” refers to “file name (and its type)”. This shows a disparency with what the authors demonstrate in Figure 5.

The presentation of this paper can be improved:
1. Figure 1 could be clearer by avoiding the overlaying of graphical items in the middle.
2. Line 243: “GPT’s internal expansion”

[1] GPTZoo: A Large-scale Dataset of GPTs for the Research Community
[2] A First Look at GPT Apps: Landscape and Vulnerability

**Questions:**

Please see my comments above.

**Ethics Review Description:**

Please see my comments above.

**Ethics Review Flag:**

Yes

**Reviewer Confidence:**

4: The reviewer is certain that the evaluation is correct and very familiar with the relevant literature

**Scope:**

3: The work is somewhat relevant to the Web and to the track, and is of narrow interest to a sub-community

---

### Official Review · Reviewer_3ZHE · 2024-12-02

**Novelty:** 4
**Technical Quality:** 5

**Review:**

The increasing adoption of customized GPT applications that leverage sensitive, domain-specific file knowledge has introduced significant security risks, as the management and protection of such sensitive data remain insufficiently explored, creating opportunities for potential data leakage vulnerabilities. This paper presents GPT-Filtor, an automated analysis framework designed to detect file knowledge leakage in GPTs at both the prompt and network transport levels. Specifically, it generates malicious prompts to assess whether GPTs disclose file knowledge, while also intercepting network traffic to analyze how this knowledge is transmitted and referenced.

**Pros:**

1. The paper is the first comprehensive analysis of file knowledge leakage in GPTs, addressing both prompt-level and network-level risks.

2. The paper has overcome barriers like anti-automation measures to provide a scalable framework for systematically evaluating the effectiveness of GPTs' protective measures against file knowledge leakage at both prompt and network levels.

**Cons:**

1. Limited to macOS: GPTs-Filtor relies on AppleScript and Charles Proxy for automation and network traffic capture, which may limit its use to macOS systems. This could restrict its applicability on other platforms or environments. Besides, while GPTs-Filtor uses AppleScript to bypass automation detection, this approach might be detected and blocked in future versions of GPTs or by OpenAI's new anti-automation mechanisms, rendering this method less effective over time.

2. Incomplete Coverage of File Exposure: The framework primarily targets harmful prompts and network traffic monitoring. However, there could be other ways that file knowledge could be exposed that are not covered by the framework.

3. The security implications of file knowledge leakage is unclear: While the paper analyzed file knowledge leaks (e.g., filenames or metadata), there's no explicit evidence that they accessed or disclosed the actual content of the files. How to exploit the leaked file knowledge is still unclear to me.

**Questions:**

1. While Section 6.3 discussed the limitations, what are the risks associated with third-party API integrations in GPTs, and how could these interactions lead to file knowledge leakage or other security issues? Pls discuss more.

2. How can we make GPTs-Filtor cross-platform (Windows, Linux, etc.), and what challenges or technical limitations might arise in doing so? Pls discuss more.

3. How might other types of input (such as code snippets, scripting languages, or hybrid commands) affect the security of GPTs, when combined with file-related prompts? Pls discuss more.

4. While Section 6.1 discussed the impact of leaked file knowledge, I am still unclear on the concept. Could you provide one or more examples demonstrating how the knowledge from the leaked file could be exploited in any one of the three attacks in Section 6.1?

**Ethics Review Description:**

Informing the developers only after the paper is accepted could delay the paper's publication and allow for vulnerabilities to remain unaddressed. Responsible disclosure before or along with the submission allows OpenAI and the developers time to take necessary protective measures, ensuring that the findings are not exploited before the system is secure.

**Ethics Review Flag:**

Yes

**Reviewer Confidence:**

3: The reviewer is confident but not certain that the evaluation is correct

**Scope:**

4: The work is relevant to the Web and to the track, and is of broad interest to the community

---

### Official Review · Reviewer_nxiM · 2024-12-03

**Novelty:** 6
**Technical Quality:** 6

**Review:**

This paper proposes GPTs-Filtor, a measurement framework for assessing the leakage of file knowledge from OpenAI's custom GPTs in terms of direct file knowledge as responses to prompts as well as indirectly via meta information in network packets. This work suggests that roughly half of the analyzed GPTs are susceptible to some leakage of file knowledge, and that roughly 3.6k file contents could be unintentionally exfiltrated.

The paper is overall well-written and highlights an important issue with custom GPTs. Even though using (external) custom GPTs in the first place should be done with some reservation regarding privacy and confidentiality, users and developers should be aware of the possibilities of unintended knowledge leakage.

The presentation of the paper is mostly adequate and I only have minor comments:

- The use of GPT (implying the main model used by ChatGPT) and GPTs (implying custom GPTs) is getting confusing throughout the paper. At some points, this convention also leads to constructs like "GPTs's" and should be reconsidered.
- Section 1: "a remarkable achievement", in the spirit of scientific objectiveness, I would argue for dropping this not-needed clause.
- Section 2.1: I do not think this section is strictly necessary for the overall paper. Maybe there is an implicit tie/underlying evolvement that I am not seeing, but then I suggest making this tie more explicit.
- Section 3.2: The example of developer-set rules is confusing; not incorrect, but
- Section 4: "the HateXplain" -> "HateXplain"?
- Section 4: "we innovative employ"; grammar issue and the innovative nature is noticeably emphasized, so I suggest removing "innovative" altogether
- Section 5: I get the need for cutting off the measurements at some points; however, for a camera-ready version of the paper I suggest just stating the cut-off date and not emphasize that it resulted from the paper submission deadline
- Section 5.2: "beave" -> behave
- Section 6.2: Hyphenation Ope-nAI should be corrected

**Questions:**

- Section 1: Crawling 8000 popular GPTs, how is popularity measured here? Could the list be compared in some way to the Alexa top lists?
- Section 1: Why only follow responsible-disclosure means upon acceptance of the paper?
- Section 5.1: Is the knowledge-base FAQ a reliable measure to determine which GPTs are relevant for GPTs-Filtor (or could developers give false information there)? Is GPTs-Filtor able to work without this information?
- Section 5.2: Are the files leaked sensitive after all, or are they, e.g., coming from the public domain?

**Reviewer Confidence:**

3: The reviewer is confident but not certain that the evaluation is correct

**Scope:**

4: The work is relevant to the Web and to the track, and is of broad interest to the community

---

### Official Review · Reviewer_PNmg · 2024-12-05

**Novelty:** 4
**Technical Quality:** 4

**Review:**

Summary
- In this paper, the authors conduct the first large-scale analysis of file knowledge leakage in third-party applications built on top of ChatGPT, known as GPTs. They develop a tool called GPTs-Filtor to automatically test and detect file knowledge leakage from GPTs at both the prompt level and the network traffic level. The authors construct a dataset of 8,000 popular GPTs across 8 categories and gather their metadata from third-party GPT app stores. GPTs-Filtor includes a specialized library of harmful prompts targeting file knowledge, which it injects into GPTs to determine if sensitive files are exposed. It also captures network traffic during interactions to extract any leaked file contents. The results show that around half of the GPTs under evaluation are vulnerable to prompt injection, and network traffic analysis reveals 3,645 leaked files from 618 GPTs. These findings highlight the need for improved security practices regarding file knowledge handling in the GPT app ecosystem.

Strengths
+ This is the first study to investigate file knowledge leakage in GPTs.
+ The authors design an automated tool, GPTs-Filtor, to detect file knowledge leakage.
+ GPTs-Filtor uncovers several cases of file knowledge leakage in real-world GPTs.

Weaknesses
- It appears that GPTs-Filtor is a combination of existing tools.
- The reliability of the results is uncertain.
- The real-world security impacts of file knowledge leakage seem limited.

Detailed comments
- Thank you for your submission. I appreciate that the authors have disclosed the file knowledge leakage problems in GPTs and developed an automated tool named GPTs-Filtor to detect them in real-world GPTs. I have a few suggestions and comments on the current version of the paper.

1. According to the content of Section 4, the harmful prompts that trick GPTs into leaking file knowledge are taken from previous studies. In my view, this weakens the contribution of this paper. Therefore, I wonder whether the authors could design more effective harmful prompts.

2. Similarly, the strategy adopted by GPTs-Filtor to simulate user interactions with OpenAI is similar to those used to bypass anti-crawlers. Therefore, I wonder if there are any differences between the proposed approach based on AppleScript and those used for anti-crawling.

3. In Section 5, the authors directly present the results of GPTs-Filtor on real-world GPTs. In my view, the authors should build a ground truth dataset (e.g., containing manually crafted GPTs) and use the GPTs in this dataset to prove the correctness and reliability of GPTs-Filtor’s results. Currently, I have concerns about whether the detected cases of file knowledge leakage are real. Is it possible that the leaked knowledge is fake, as GPTs may have adopted defensive mechanisms?

4. In Section 6.1, the authors merely speculate on several security impacts caused by file knowledge leakage. Is it possible for the authors to present a few real security impacts caused by the leaked file knowledge found in real-world GPTs? For example, in the Abstract, the authors mention that there may be sensitive information in the leaked file knowledge. Is it possible to inspect the leaked file knowledge to determine if any sensitive information has been leaked?

**Questions:**

Please see the detailed comments above.

**Reviewer Confidence:**

4: The reviewer is certain that the evaluation is correct and very familiar with the relevant literature

**Scope:**

4: The work is relevant to the Web and to the track, and is of broad interest to the community